# VISION-SR1: SELF-REWARDING VISION-LANGUAGE MODEL VIA REASONING DECOMPOSITION AND MULTI-REWARD POLICY OPTIMIZATION

**Zongxia Li**[1,2†], **Wenhao Yu**[1†], **Zhenwen Liang**[1], **Chengsong Huang**[1,3], **Rui Liu**[1,2],
**Fuxiao Liu**[2], **Jingxi Chen**[2], **Dian Yu**[1], **Jordan Boyd-Graber**[2], **Haitao Mi**[1], **Dong Yu**[1]

[1]Tencent AI Seattle Lab, [2]University of Maryland, College Park,
[3]Washington University in St. Louis
† Core contributors
zli12321@umd.edu; wenhaowyu@global.tencent.com; jordanbg@gmail.com

## ABSTRACT

Vision-Language Models (VLMs) often suffer from visual hallucinations – generating things that are not consistent with visual inputs – and language shortcuts, where they skip the visual part and just rely on text priors. These issues arise because most post-training methods for VLMs rely on simple verifiable answer matching and supervise only final outputs, leaving intermediate visual reasoning without explicit guidance. As a result, VLMs receive sparse visual signals and often learn to prioritize language-based reasoning over visual perception. To mitigate this, some existing methods add visual supervision using human annotations or distilled labels from external large models. However, human annotations are labor-intensive and costly, and external signals can introduce high latency cost.

In this paper, we introduce Vision-SR1, a three-stage self-rewarding reinforcement learning method that improves visual reasoning without relying on external visual supervision. Vision-SR1 decomposes VLM reasoning into two components: *visual reasoning* and *language reasoning*, where the model is first prompted to produce self-contained visual descriptions sufficient to answer the question without referring back to the input image, before jointly optimizing both visual and language reasoning through our multi-reward loss objective. To validate this self-containment, the same VLM model is re-prompted to perform language reasoning using only the generated visual reasoning as input to compute visual reward. The final reward is computed through a decoupled reward-advantage framework, where visual reward and language reasoning reward each have their advantages, log probabilities, and KL divergence calculated separately. This decoupling enables more fine-grained reward computation by preventing the entanglement of heterogeneous reward signals. Our experiments show that Vision-SR1 improves visual reasoning, mitigates visual hallucinations, and reduces reliance on language shortcuts across diverse vision-language tasks, while being more efficient than methods that rely on external visual reward models, which require additional GPUs to host. In contrast, Vision-SR1 introduces no extra GPU overhead beyond that of standard training.

○ Code: https://github.com/zli12321/Vision-SR1.

## 1 INTRODUCTION

Recent advances in vision-language models (VLMs) have progressed by integrating pre-trained language models and vision encoders with instruction tuning (Liu et al., 2023b; et al, 2024; Chen et al., 2024; Bai et al., 2025; Li et al., 2025d). Despite these successes, a critical limitation remains in their reasoning capabilities: VLMs often produce visual hallucinations—descriptions of content that is not actually present in the image (Guan et al., 2024; Liu et al., 2024; Li et al., 2025e; Liu et al., 2023a)—or rely on language shortcuts, where the model bypasses visual understanding and instead

depends solely on text priors (Si et al., 2022; Bleeker et al., 2024). R1-style reinforcement learning (RL) methods have recently improved the reasoning abilities of VLMs across diverse tasks (Huang et al., 2025b; Shen et al., 2025; Xia et al., 2025; Zhang et al., 2025). However, these methods often encourage "thinking over seeing", leaning heavily on language reasoning while demoting visual perception (Liu et al., 2025; Yao et al., 2025). This imbalance makes VLMs susceptible to reward hacking (Fu et al., 2025) and spurious effects (Shao et al., 2025) in RL training. Although VLMs trained with RL often "improve", these improvements often just are probability shifts toward the style of training and test data, leading to language shortcut answers from prior knowledge and overlooking hallucination risks (Li et al., 2025b).

In essence, most existing post-training methods for VLMs rely on a one stage, simple, verifiable answer matching and thus lack explicit supervision for visual information. As a result, VLM's visual signals are sparse, leading them to prioritize language-based reasoning over visual perception. To mitigate this, some methods introduce intermediate visual supervision through human annotations (Thawakar et al., 2025) or distilled labels (e.g., pre-extracted key steps) from external models (Xu et al., 2024; Zhang et al., 2025; Xiao et al., 2025; Xia et al., 2025; Lu et al., 2025). However, these solutions have significant limitations. Human annotations are labor-intensive, costly, and difficult to scale across multimodal tasks, while distilled signals inherit biases and latency from source models and often fail to generalize across diverse domains. Moreover, distributional shifts between fixed intermediate signals and the continually updated policy can lead to reward hacking (Gao et al., 2023). Most importantly, both approaches remain limited by their reliance on external supervision or simply summimg up multiple intermediate rewards, restricting their scalability and applicability.

This paper introduces **Vision-SR1**, a reinforcement learning framework that encourages VLMs to produce *self-contained* visual reasoning that can be verified by the VLM itself without external supervision. Vision-SR1 explicitly isolates the visual grounding stage from the reasoning stage: *visual perception* and *language reasoning*. The visual perception is required to capture all details relevant to answering the query, so that the reasoning stage can proceed without re-accessing the original image. We explicitly compute advantages and rollouts separately for each stage, then calculate individual Actor policy losses and KL divergence terms for the visual perception and language reasoning stages before combining them into a unified training objective.

The training has two **rollout passes** and one **training objective optimization** of the same VLM:

– **First pass (standard rollout):** $(\text{Image}, \text{Query}) \rightarrow (\text{Visual Perception}, \text{CoT Reasoning}, \text{Answer})$

- The model generates a structured output that explicitly separates visual perception, chain-of-thought (CoT) reasoning, and the final answer.
- An **accuracy reward** compares the final answer with the ground truth.

– **Second pass (self-reward rollout):** $(\text{Query}, \text{Visual Perception}) \rightarrow (\text{CoT Reasoning}, \text{Answer})$

- The model is re-prompted to reason using only the generated perception (without re-accessing the original image). If the correct answer is derived, the perception is considered **faithful**, and a **self-visual reward** is assigned.

– **Multi-Reward Policy Optimization (objective optimization):**

- The multi-reward policy optimization enables the policy model to receive distinct feedback for visual reasoning quality and answer accuracy through separate advantage computations and rollout-specific loss terms.

The self-rewarding process eliminates the computational overhead of deploying additional reward models on separate GPUs, while the decoupled reward signals are combined through our multi-policy loss objective to provide balanced training that strengthens both visual perception and language reasoning without the entangled learning signals of traditional reward summation. Vision-SR1 improves visual reasoning, mitigates hallucinations, and reduces language shortcuts across diverse vision-language tasks.

## 2 METHOD

We build on Group Relative Policy Optimization (Shao et al., 2024, GRPO) for improving VLM reasoning. We first review the key concepts then introduce our method.

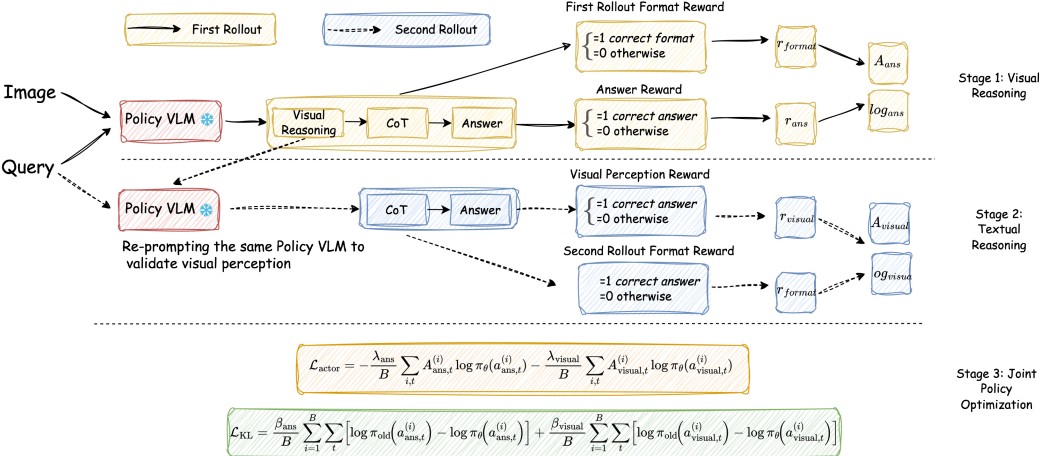

Figure 1: Overall framework of Vision-SR1. During RL training the VLM has two rollouts. In the first pass, the model takes an image–query pair and generates a structured output (visual perception, CoT reasoning, and answer), with answer reward computed against the ground truth. In the second pass, the model is re-prompted to answer using only query and its generated visual perception. If the correct answer is derived, a self-visual reward is assigned. We compute the advantages and log probabilities for each rollout for Multi-Reward Policy Optimization.

## 2.1 PRELIMINARY: REINFORCEMENT LEARNING FOR VLM WITH VERIFIABLE REWARD

We optimize a pre-trained VLM as a policy $\pi$ to be optimized in reinforcement learning. Given a multimodal question ($Q$) with an image $i$ and a text question $q$, where $Q = \{i, q\}$, the policy model $\pi$ generates a reasoning response $s$. GRPO optimizes the response $s$ for the policy model. For each multimodal question $Q = \{i, q\}$ we sample a *group* of $K$ candidate responses $\mathcal{S}_Q = \{s_1, \ldots, s_K\}$, $s_k \sim \pi_\theta(\cdot | Q)$. Each response is scored by a scalar reward $r(Q, s_k)$ (defined in Sec. 2.2), and we compute a *group-relative* advantage

$$\hat{A}^{\mathrm{grp}}(Q, s_k) = r(Q, s_k) - \frac{1}{K} \sum_{j=1}^{K} r(Q, s_j), \tag{1}$$

which centers rewards within the group, removing question-level biases while retaining pairwise preferences. We update the policy by maximizing

$$\mathcal{L}_{\mathrm{GRPO}}(\theta) = \mathbb{E}_{Q \sim \mathcal{D}} \left[ \sum_{k=1}^{K} \hat{A}^{\mathrm{grp}}(Q, s_k) \log \pi_\theta(s_k | Q) - \beta \, \mathrm{KL}\big(\pi_\theta(\cdot | Q) \| \pi_{\theta_0}(\cdot | Q)\big) \right], \tag{2}$$

where $\pi_{\theta_0}$ is the frozen, pre-trained reference model and $\beta$ controls the strength of the KL penalty that keeps the updated policy close to its original behavior.

The group-centred baseline in equation 1 guarantees $\sum_k \hat{A}^{\mathrm{grp}}(Q, s_k) = 0$, thereby reducing the variance of policy-gradient estimates without requiring an external value critic.

## 2.2 STAGE 1 & 2: SELF-REWARDING VLM VIA REASONING DECOMPOSITION

Incorporating intermediate visual supervision can strengthen the reasoning ability of VLMs. However, existing methods suffer from key limitations: approaches based on human annotations are labor-intensive and costly (Thawakar et al., 2025), while those that distill supervision from external models introduce additional compute overhead and training latency (Zhang et al., 2025; Xiao et al., 2025; Xia et al., 2025). To overcome these issues, we introduce a self-reward framework that enables the VLM to reward its own visual reasoning. The key idea is to decompose the visual reasoning process into structured components, i.e., the VLM first produces a self-contained visual perception and then assesses whether this perception is sufficient for produce the final answer. This decomposition reduces reliance on external supervision and allows the reward signal to adapt dynamically as the model improves.

**Decomposing VLM Reasoning.** To encourage the VLM to reason about its visual input, we require every response to adhere to a ***See-Think*** generation format (Jia et al., 2024; Xia et al., 2025) format. Specifically, for a vision-language task, $Q = \{i, q\}$ where $i$ is the input image and $q$ is the textual query, the model produces the following structured output:

$$\langle \text{visual\_reasoning} \rangle \, c \, \langle /\text{visual\_reasoning} \rangle \, \| \, \langle \text{think} \rangle \, t \, \langle /\text{think} \rangle \, \| \, \langle \text{answer} \rangle \, a \, \langle /\text{answer} \rangle$$

where $c$ is a *self-contained* visual reasoning that captures all visual information necessary to solve the task, so that the following language reasoning can proceed without re-accessing the original input image. Besides, $t$ is the language reasoning trace, and $a$ is the final answer.

**Self-Reward for Visual Reasoning.** A challenge is judging whether the visual reasoning $c$ is *self-contained*: i.e., whether it encodes *all* the visual information needed to answer the question $Q = \{i, q\}$ correctly. Our idea is to treat the visual perception as a *text-only proxy* for the image and validate it by re-prompting the VLM itself to perform language reasoning using only the generated perception as input. If the model can derive the correct answer from $(c, q)$ alone, we consider $c$ to be visually faithful and assign the corresponding visual reward.

$$\hat{a} \; = \; f_\theta(c, q), \quad r_{\text{visual}}(Q, c) \; = \; \mathbb{I}[\hat{a} = a^*], \tag{3}$$

where $a^*$ is the ground-truth answer. Instead of using an external reward model, we leverage the policy model's own reasoning ability for self-evaluation. The model itself determines the reward by answering the question using only its generated visual reasoning (Figure 1).

**Reward Composition.** The reward combines three *aligned* components, each conditioned on the question $Q = \{i, q\}$:

● **Format reward** $r_{\text{fmt}}(s)$: measures whether the response strictly follows the required layout. This reward is applied to both Visual reward and Accuracy reward.

● **Answer reward** $r_{\text{ans}}(Q, a)$: measures the correctness of the final answer ($r_{acc}$) *plus* the corresponding format reward. Because $a$ is generated after the reasoning trace $t$, the term implicitly rewards CoT reasoning. This is computed at first rollout with hyper-parameters ($0 \leq \alpha \leq 1$):

$$r_{ans}(Q, a) = \; r_{\text{acc}}(Q, a), + \alpha \, r_{\text{fmt}}(s) \tag{4}$$

● **Visual reward** $r_{\text{visual}}(Q, c)$: measures whether the visual reasoning output is self-contained, i.e., sufficient to answer the question without image ($r_{vis\_acc}$) *plus* corresponding format reward. A reward of 1 is assigned if, given only the question and the visual reasoning, the VLM can give the correct answer. This is computed at second rollout:

$$r_{visual}(Q, a) = \; r_{\text{vis\_acc}}(Q, a), + \alpha \, r_{\text{fmt}}(s) \tag{5}$$

### 2.3 Stage 3: Multi-Reward Optimization with Multi-Advantage Loss Computation.

Simply *summing* the visual reasoning reward and the final-answer accuracy reward could produce a *sparse and entangled* learning signal: the policy has little to tell which rollout was responsible. To disentangle visual reasoning and answer accuracy assignment, we keep the two rollouts—answer generation and visual reasoning—*separate* throughout the update. Each rollout receives its own log-probabilities, advantage, and KL term, and the gradients are combined only at the end. This turns the single *multi-reward* problem into two single-reward sub-problems that share parameters with individually optimized feedback.

**Reward-Specific Log-Probability Tracking.** During sampling we cache the behavioral log probabilities for every token in each rollout:

$$\log \pi_{\text{old}}^{(i)}(a_{\text{ans}, t}), \; \log \pi_{\text{old}}^{(i)}(a_{\text{visual}, t}),$$

where $a_{\text{ans}, t}$ is the action at step $t$ of the first rollout, and $a_{\text{visual}, t}$ is the action (token) at step $t$ of the second rollout. At update time we compute the corresponding $\log \pi_\theta$ under the current parameters to compute the policy and KL losses.

**Group-wise $z$-Score Advantage.** For each reward we follow GRPO to compute the advantage:

$$A_{\text{ans}}^{(i)} = \frac{r_{\text{ans}}^{(i)} - \mu_{\text{ans}}}{\sigma_{\text{ans}} + \varepsilon}, \qquad A_{\text{visual}}^{(i)} = \frac{r_{\text{visual}}^{(i)} - \mu_{\text{visual}}}{\sigma_{\text{visual}} + \varepsilon}, \tag{6}$$

with means and standard deviations $\mu_{\text{ans}} = \frac{1}{B}\sum_i r_{\text{ans}}^{(i)}$, $\sigma_{\text{ans}}^2 = \frac{1}{B}\sum_i \left(r_{\text{ans}}^{(i)} - \mu_{\text{ans}}\right)^2$, where $B$ is the rollout batch size (and analogously for the visual group). Broadcasting $A_{\text{ans}}$ to all caption tokens and $A_{\text{visual}}$ to all answer tokens gives two advantage masks that weight the corresponding log-probabilities during backpropagation for each sub-task.

**Actor Loss (Policy Gradient Loss).** The actor loss computes weighted policy gradients for the two reward signals (answer and visual), with separate coefficients $\lambda_{\text{ans}}$ and $\lambda_{\text{visual}}$ indicating their contributions.[1]

$$\mathcal{L}_{\text{actor}} = -\frac{1}{2B}\sum_{i,t}\left(A_{\text{ans},t}^{(i)}\log\pi_\theta(a_{\text{ans},t}^{(i)}) + A_{\text{visual},t}^{(i)}\log\pi_\theta(a_{\text{visual},t}^{(i)})\right) \tag{7}$$

, where $B$ is the rollout batch size (as in Eq. 6).

**KL Divergence Regularization Loss.** The KL regularization applies separate penalty coefficients $\beta_{\text{cap}}$ and $\beta_{\text{ans}}$ to prevent excessive policy deviation for each reward component.

$$\mathcal{L}_{\text{KL}} = \frac{\beta_{\text{ans}}}{B}\sum_{i=1}^{B}\sum_t\left[\log\pi_{\text{old}}(a_{\text{ans},t}^{(i)}) - \log\pi_\theta(a_{\text{ans},t}^{(i)})\right] + \frac{\beta_{\text{visual}}}{B}\sum_{i=1}^{B}\sum_t\left[\log\pi_{\text{old}}(a_{\text{visual},t}^{(i)}) - \log\pi_\theta(a_{\text{visual},t}^{(i)})\right]$$

$$\tag{8}$$

**Multi-Reward Loss Objective.** The total loss combines multi-reward policy gradients with component-specific regularization to optimize the model across both reward signals.

$$\mathcal{L}_{\text{total}} = \mathcal{L}_{\text{actor}} + \mathcal{L}_{\text{KL}} \tag{9}$$

## 2.4 THEORETICAL ANALYSIS

We analyze why Multi-Reward Policy Optimization with separate advantage computation could improve VLM RL training compared to using only answer rewards. In standard RL training, the objective depends solely on final answer correctness:

$$\nabla_\theta \mathbb{E}_{s\sim\pi_\theta}[r_{\text{ans}}(a, a^*)] \tag{10}$$

where $s = (t, a)$ contains visual reasoning and language reasoning trace $t$ and final answer $a$. Since $r_{\text{ans}}$ only measures whether $a$ matches ground truth $a^*$, the intermediate visual reasoning $t$ receives no direct supervision signal. For VLMs, the stronger LLM backbone dominates generation of $t$, and continued RL training leads to potential reward hacking where the model exploits language priors to achieve correct answers without visual grounding Pantazopoulos & Özyiğit (2025).

**Multi-Reward Loss Decomposition.** We decompose the loss computation itself into separate components as shown in Equation 9, where the actor loss handles visual and answer components separately (with the $KL$ regularization term following similar component-wise structure):

$$\mathcal{L}_{\text{actor}} = -\lambda_{\text{ans}}\mathbb{E}[A_{\text{ans}}\log\pi_\theta(a_{\text{ans}})] - \lambda_{\text{visual}}\mathbb{E}[A_{\text{visual}}\log\pi_\theta(a_{\text{visual}})] \tag{11}$$

Since each advantage is computed from different reward components (visual and answer, in Equation 6), this approach creates clear gradient paths from each reward to its corresponding components, enabling independent optimization of visual reasoning and language reasoning capabilities (Zhu et al., 2025; Lyu et al., 2025; Liu et al., 2026).

---

[1]We use 0.5 for $\lambda_{\text{ans}}$ and $\lambda_{\text{visual}}$.

### 2.4.1 COMPUTATIONAL EFFICIENCY ANALYSIS

A natural question with two-stage rollout training is whether it doubles the computational cost relative to standard one-stage GRPO. This section's analysis shows two-stage rollout is only 10-15% more expensive than standard GRPO, while requiring no extra GPU computation.

In training, both rollout stages reuse the same VLM without loading additional models nor calling external APIs. In practice, two-stage rollout training adds only a modest overhead over standard GRPO. For example, training for 20 steps (per-device batch size 8, 8 GPUs) on a 7B model takes standard GRPO approximately 10.5 hours, while our two-stage training takes approximately 13 hours, an overhead of roughly 20%. Additionally, we compare against two alternative external visual reward strategies. 1): Using proprietary models (GPT-5 Singh et al. (2025), Claude Anthropic, Gemini Peng et al. (2025b)) as an external judge requires evaluating $N \times B$ responses per step, where $N$ is the number of rollouts per sample and $B$ is the effective batch size, which can cause API rate limits are hit quickly, pushing training time more than doubling the cost. Using a local open-source judge instead requires dedicating at least one GPU to inference, reducing training parallelism from $N$ to $N - 1$ GPUs and introducing additional inference latency. By contrast, self-reward training avoids both penalties, making two-stage training practical and scalable without incurring the overhead of external judges.

## 2.5 DATA PREPARATION

**Vision-SR1-47K.** Our RL dataset consists of approximately 47K examples collected from 24 open-source VLM benchmarks. It spans three key reasoning domains (Figure 1): mathematical reasoning (30.5%), which strengthens quantitative and logical abilities; commonsense knowledge (30%); and general visual understanding (39.5%), which grounds the model in visual question answering.

## 3 EXPERIMENTS

To implement our Vision-SR1, we use **Qwen2.5-VL-3B** and **7B**, **Mimo-7B-VL** Team et al. (2025) as base models. We train the base model with GRPO. The RL phase is trained for 200 steps on the Vision-SR1-47K dataset. During training, the policy model first generates visual reasoning from the input image, then produces language reasoning and final answer. We then compute a self-reward for visual reasoning by re-prompting the frozen policy model to answer the question using only its generated visual reasoning, without access to the original image $i$. Finally, we compute advantages and log probabilities separately for each reward component and combine them in the final loss (Figure 1).[2]

Table 1: Vision-SR1-47K data comprises three domains—Math, Knowledge, and General Visual Reasoning—providing diverse supervision for VLM generalization and adaptation.

| Category | Included Datasets | Size | (%) |
|---|---|---|---|
| Math | CLEVR-Math, GeoQA+, UniGeo, GEOS, Geometry3K, Super-CLEVR | 14K | 30.5% |
| Science Knowledge | TQA, ScienceQA, AI2D, PMC-VQA, VQA-RAD, EXAMS-V-train | 14K | 30% |
| General Visual Reasoning | ChartQA, DVQA, PlotQA, FigureQA, MapQA, TabMWP, A-OKVQA, IconQA, visual7w, OpenSpaces, Spacellava | 18K | 39.5% |

## 3.1 BASELINE METHODS

**Vision-R1** (Huang et al., 2025b): The first R1-style reinforcement learning approach, which relies solely on answer rewards as the training signal. However, since the original Vision-R1 was trained only on the math domain and falters on general-domain reasoning, we reproduce it using our 47K dataset to ensure a fair comparison.

**Perception-R1** (Xiao et al., 2025)'s training resembles Vision-R1 but incorporates pre-extracted visual annotations as an additional reward signal. These visual annotations are derived from a state-of-the-art proprietary multimodal LLM (not specified in the paper).

---

[2]The policy model remains frozen during both rollouts.

Table 2: Vision-SR1 vs. baselines. For Vision-R1, as noted in Section 3.1, the original model checkpoint was trained only on math-domain data. So we also reproduce it using our 47K dataset.

| Methods | General Visual Understanding | | | | Visual Math & Hallucination | | | Avg. |
|---|---|---|---|---|---|---|---|---|
| | MMMU -Pro | MMMU | RealWorld QA | VisNum Bench | Math Verse | MATH -Vision | Hallusion Bench | |
| Visionary-R1 (3B) by Xia et al. (2025) | 27.4 | 30.6 | 56.9 | 10.0 | 45.0 | 40.4 | 26.7 | 33.9 |
| Percention-R1 (7B) by Xiao et al. (2025) | 36.8 | 40.9 | 69.4 | 15.9 | 52.1 | 35.7 | 65.4 | 45.2 |
| Vision-R1 (7B) by Huang et al. (2025b) | 34.9 | 42.8 | 60.1 | 33.0 | 57.3 | 51.2 | 32.2 | 44.5 |
| *Backbone model: Qwen2.5-VL-3B* | | | | | | | | |
| Zero-shot Inference (before RL) | 30.5 | 25.5 | 65.4 | 15.7 | 44.3 | 40.4 | 27.1 | 35.5 |
| Vision-R1 47K data (fair comparison) | 40.3 | 49.5 | 63.0 | 36.7 | 42.8 | 29.9 | 67.4 | 47.1 |
| **Vision-SR1 (ours)** | 40.8 | 49.6 | 66.1 | 41.9 | 45.8 | 29.3 | 68.3 | 48.8 |
| *Backbone model: Qwen2.5-VL-7B* | | | | | | | | |
| Zero-shot Inference (before RL) | 34.2 | 33.5 | 68.5 | 21.4 | 49.2 | 31.9 | 51.7 | 41.5 |
| Vision-R1 47K data (fair comparison) | 39.8 | 51.8 | 66.6 | 43 | 53.2 | 33.8 | 66.6 | 50.7 |
| **Vision-SR1 (ours)** | 40.7 | 52.2 | 69.2 | 43.5 | 54.5 | 36.2 | 68.9 | 52.2 |
| *Backbone model: Mimo-VL-7B* | | | | | | | | |
| Zero-shot Inference (before RL) | 38.0 | 45.6 | 68.2 | 30.2 | 35.5 | 21.6 | 71.9 | 44.4 |
| Vision-R1 47K data (fair comparison) | 38.7 | 47.3 | 67.1 | 33.5 | 35.3 | 25.7 | 74.3 | 46.0 |
| **Vision-SR1 (ours)** | 39.3 | 49.5 | 68.1 | 44.6 | 40.0 | 29.6 | 75.6 | 49.5 |

**Visionary-R1** (Xia et al., 2025): Trained to produce a caption–reason–answer output format during RL, where the supervision signal comes from an external text-only LLM (not specified in the paper).

For fair comparisons, we only re-train Vision-R1 on our 47K dataset, since both Perception-R1 and Visionary-R1 require access to external annotations or supervision signals, which are undisclosed.

## 3.2 BENCHMARKS AND METRICS

Our evaluation covers three areas to evaluate VLMs abilities. Specifically, the domains include (1) general visual understanding, (2) multimodal math reasoning (3) visual hallucination detection.

**General Visual Understanding.** We evaluate general visual understanding across five diverse benchmarks. **MMMU** (Yue et al., 2024) tests cross-modal reasoning and subject knowledge with 11.5K college-level, four-choice questions spanning six disciplines. **MMMU-Pro** (Yue et al., 2025) increases the difficulty with ten choices per question and adds a challenging *vision-only* setting, where all text is embedded within the image to necessitate robust visual parsing. **Real-WorldQA** (xAI, 2024) features ∼700 real-world images from vehicle captures, paired with spatially grounded questions that require verifiable answers. **VisNumBench** (Weng et al., 2025) specifically targets visual number sense through ∼1.9K questions covering seven numerical attributes and four estimation tasks.

**Multimodal Mathematical Reasoning.** We assess mathematical reasoning using two specialized benchmarks. **MathVerse** (Zhang et al., 2024a) consists of 2.6K diagram-centric problems (e.g., geometry, functions), each rendered in six visual-text variants to disentangle true visual understanding from linguistic shortcuts. Evaluation is based on step-by-step Chain-of-Thought (CoT) correctness. **MATH-Vision** (Wang et al., 2024) presents ∼3K competition-grade problems across 16 disciplines and five difficulty levels, stressing advanced multimodal reasoning.

**Hallucination Diagnosis.** To diagnose model failures, we use **HallusionBench** (Guan et al., 2024), a benchmark designed to pinpoint specific errors: (i) language-side hallucination, where visual context is ignored, and (ii) visual-illusion errors, where the image is misinterpreted. Because the benchmark uses binary yes/no questions with unambiguous ground truth, each incorrect response can be cleanly attributed to one of these two failure modes.

For our evaluations, we all use Gemini-2.5-flash (Comanici et al., 2025) to judge response correctness on non-multiple choice format questions, serving as a proxy for human judgment.

## 3.3 EXPERIMENTAL RESULTS

### 3.3.1 VISION-SR1 V.S. BASELINE METHODS

Table 2 compares Vision-SR1 with several baseline methods across diverse vision-language benchmarks. With the Qwen2.5VL-7B backbone, Vision-SR1 reaches 40.7 on MMMU-Pro and 52.2 on

MMMU, outperforming Vision-R1 fair comparison runs (34.9 and 42.8, respectively). When averaged across all benchmarks, Vision-SR1 establishes a clear margin of improvement. With the 72B backbone, it achieves an average score of 52.2, compared to 44.5 for Vision-R1. Even with the smaller 3B backbone, Vision-SR1 achieves 48.8 average, outperforming all comparable baselines. For results on Mimi-VL-7B, a model outside the Qwen-VL family, we observe a similar trend: the average accuracy improves from 44.4 to 49.5. This shows that our method generalizes beyond the Qwen-VL. These results demonstrate that Vision-SR1 outperforms prior baseline models across both general-purpose and math-specific visual reasoning tasks, validating the effectiveness of our approach.

### 3.3.2 Ablation study on Spatial Reasoning and Language Shortcut Datasets

In addition to evaluating Vision-SR1 on standard visual-reasoning benchmarks, we further evaluate its effectiveness on additional datasets to probe two complementary challenges: spatial reasoning and language-shortcut (LS) robustness. MMSI-Bench Yang et al. (2025) and OmniSpatial Jia et al. (2025) target multi-image spatial understanding, requiring models to integrate spatial relationships across multiple images. In contrast, ViLP Luo et al. (2025) evaluates language shortcuts by pairing each question with images that can be answered either through textual priors alone or only through

Table 3: Our method also can improve VLMs' abilities on spatial reasoning and language shortcut (LS) robustness.

| Methods | ViLP (LS) | MMSI -Bench | Omni Spatial | Avg. |
|---|---|---|---|---|
| *Backbone: Mimo-VL-7B* | | | | |
| before RL | 56.4 | 28.2 | 40.3 | 41.6 |
| Vision-R1 | 58.2 | 27.7 | 40.4 | 42.1 |
| **Vision-SR1** | 59.3 | 28.0 | 42.7 | 43.3 |
| *Backbone: Qwen2.5-VL-7B* | | | | |
| before RL | 45.1 | 24.0 | 27.3 | 32.1 |
| Vision-R1 | 51.3 | 21.9 | 31.1 | 34.8 |
| **Vision-SR1** | 52.6 | 27.7 | 44.2 | 41.5 |

pure visual reasoning. Table 3 shows that Vision-SR1 generalizes well to spatial reasoning benchmarks and substantially improves robustness to visual–language shortcuts. In particular, explicitly generating visual descriptions helps the model avoid shortcut behavior and rely more on the actual visual content. Next we propose a systematic way to evaluate VLMs' language shortcut frequencies on standard VLM benchmarks.

### 3.3.3 Analysis on Language Shortcut

Table 4: Language Shortcut Rate (LSR) across different benchmarks. Lower values indicate better performance, as a reduced LSR reflects fewer language shortcuts during reasoning. Adding additional reward supervision can reduce the change of visual reasoning reward hacking.

| Methods | General Visual Understanding | | | | Visual Math & Hallucination | | | |
|---|---|---|---|---|---|---|---|---|
| | MMMU -Pro | MMMU | RealWorld QA | VisNum Bench | Math Verse | MATH -Vision | Hallucination Bench | Avg. |
| Vision-SR1 (3B) | 7.5 | 6.3 | 10.8 | 5.4 | 10.3 | 8.3 | 10.1 | 9.4 |
| ⊢ w/o self-reward | 9.0 | 9.6 | 11.9 | 4.2 | 11.4 | 9.2 | 8.5 | 10.4 |
| Vision-SR1 (7B) | 8.0 | 6.5 | 13.4 | 4.2 | 11.5 | 10.7 | 6.8 | 9.8 |
| ⊢ w/o self-reward | 8.7 | 5.3 | 10.8 | 3.9 | 12.7 | 10.7 | 9.1 | 10.1 |

We also introduce the Language Shortcut Rate (LSR), a metric designed to quantify how often a model produces the correct answer with an incorrect visual perception. A high LSR suggests the model is leveraging language knowledge prior rather than genuine visual understanding.

Our two-step evaluation uses Gemini-2.5-flash as a judge: (1) Visual Perception Extraction: for each model output, we extracted the generated visual reasoning, denoted as $\hat{C}$. (2) Self-Containment Check: we then provide the $\hat{C}$ and the original question $Q$ to Gemini-2.5-Flash evaluator. If the evaluator can reproduce the correct ground-truth answer using *only* this information, $\hat{C}$ is self-contained. Based on this process, we define the **Language Shortcut Rate (LSR)** as the percentage of instances where the model produces an *incorrect (not self-contained) visual reasoning* but still gives the *correct final answer*:

$$\text{LSR} = \frac{\#\{\text{incorrect visual reasoning \& correct answer}\}}{\#\{\text{total samples}\}}$$

A higher LSR indicates that the model is answering correctly while bypassing visual perception, suggesting reliance on language prior shortcuts. An LSR of 0 indicates no shortcutting, i.e., every correct answer is supported by a correct, self-contained visual reasoning.

We compute the LSR for 7B model w/ and w/o self rewards on seven selected benchmarks for demo example in Table 4. Visual shortcuts *pervade multimodal mathematical reasoning*, which raises important questions in previous work R1-VL (Zhang et al., 2025), VLM-R1 (Shen et al., 2025), Vision-R1 (Huang et al., 2025b): is multimodal RL training truly improving VLMs' abilities to perform visual reasoning or simply awakening the models' language reasoning ability to guess without actually looking at visual information?

### 3.3.4 ANALYSIS OF VISUAL ATTENTION CHANGE

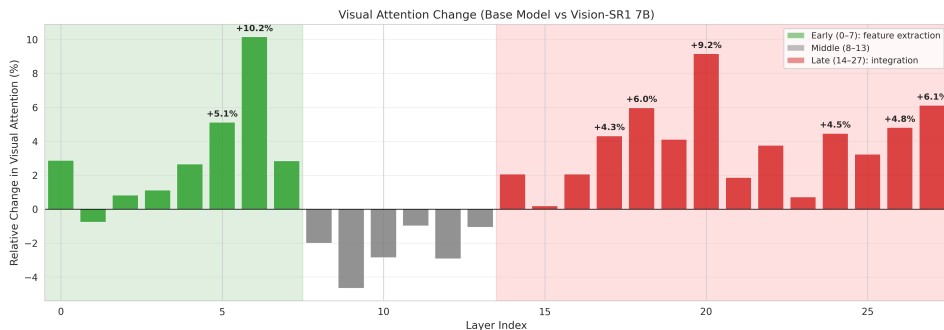

Figure 2: Post-training encourages the ViT to devote more attention to visual tokens at both early feature extraction and late integration stages, while compressing intermediate processing.

We analyze layer-wise shifts in visual attention between the base model and Vision-SR1 to understand how post-training affects visual processing. We sample 1,000 images from the ViLP dataset and compute, for each image, the per-layer difference in visual attention weights between the post-trained and pretrained models via L2 distance. Averaging these differences across all 50 samples yields a layer-wise breakdown of attention redistribution over visual tokens. As shown in Figure 2, the changes cluster into two distinct patterns. In the early layers (0–7), post-training increases visual attention, with the largest gain at Layer 6 (+10.2%), showing the model learns to extract richer low-level visual features earlier in the network. The middle layers' (8–13) decrease indicates a redistribution rather than a uniform increase in attention. In the late layers (14–27), visual attention rises again, peaking at Layer 20 (+9.2%), which shows enhanced visual re-engagement during the visual integration and output generation stages.

## 4 RELATED WORK

### 4.1 POST-TRAINING VISION-LANGUAGE MODELS

Recent vision-language models have increasingly leveraged post-training alignment techniques, including instruction tuning and reinforcement learning, to enhance general-purpose multimodal performance (Liu et al., 2023b; Bai et al., 2025; Chen et al., 2024; et al, 2024; Huang et al., 2025b). For example, LLaVA (Liu et al., 2023b) is tuned on GPT-4 generated (image, question, answer) pairs, coupling a CLIP encoder with Vicuna to produce a visual chat assistant that imitates some GPT-4 vision capabilities. InstructBLIP (Dai et al., 2023) introduces an instruction-aware query transformer tuned on 26 datasets, which yields a model that substantially outperforms even larger models on zero-shot benchmarks. Beyond standard instruction-tuning methods like LLaVA and Instruct-BLIP, recent work increasingly uses reinforcement learning (RL) to align vision-language models for better reasoning (Huang et al., 2025b; Xia et al., 2025; Xiao et al., 2025). Many of these methods, inspired by techniques from DeepSeek-R1 (DeepSeek-AI et al., 2025), focus on sophisticated reward engineering. Strategies include providing step-wise rewards to supervise the intermediate reasoning (Zhang et al., 2025), adding explicit visual annotations to ground truth for calculating visual rewards (Xiao et al., 2025), and applying RL in a two-stage curriculum that first strengthens

text-only reasoning (Peng et al., 2025b). As a complementary approach, RL from AI Feedback for VLMs demonstrates that preference-based alignment is also a powerful signal, showing it can substantially reduce object hallucination by learning from AI-generated feedback (Yu et al., 2024).

## 4.2 SELF-REWARDING REINFORCEMENT LEARNING

The existing reinforcement learning with verifiable rewards (RLVR) methods heavily rely on high-quality reward models or human feedback, creating a major bottleneck for scalability (Peng et al., 2025a; Dai et al., 2025; Li et al., 2025c; Luu et al., 2025). To overcome this, recent work explores self-rewarding approaches, where the model itself provides intrinsic reward signals during RL post-training, an idea first pioneered by Yuan et al. (2025). Building on self-rewarding language models, methods replace external reward models with the model's own confidence and uncertainty (logit-based self-certainty) or self-verification of its solutions, and even elicit a latent *endogenous* reward already present inside base LLMs (Zhao et al., 2025; Li et al., 2025a; Simonds et al., 2025; Zheng et al., 2025; van Niekerk et al., 2025; Huang et al., 2025a; Zhou et al., 2025). For example, RLIF leverages self-certainty as a reward, achieving comparable performance to GRPO while improving out-of-distribution generalization (Zhao et al., 2025). Similarly, RLSC optimizes a self-confidence reward to secure large accuracy gains with only a few training samples (Li et al., 2025a).

Although self-generated reward signals have thrived in text-only LLMs, only a few works extend this idea to VLMs (Zhou et al., 2024; Lee et al., 2025; Holmes & Chi, 2025), largely due to the complexity of the visual modality and the difficulty of properly defining and evaluating reward signals that capture visual perception. Recent progress includes Calibrated Self-Rewarding, which iteratively generates candidates, self-scores them with step-wise, visually constrained rewards, and fine-tunes via direct preference optimization (DPO) (Zhou et al., 2024). Similarly, RG-VLM uses a VLM to directly label rewards for offline trajectories in long-horizon visual tasks, serving as an auxiliary signal that boosts generalization (Lee et al., 2025). Beyond judgment-based signals, ARES derives dense shaped rewards from attention weights to accelerate learning under sparse or delayed feedback (Holmes & Chi, 2025). These works show that internal visual signals can provide rich reward feedback for VLM alignment without costly supervision, yet the reward is not integrated end-to-end, where the policy receives both visual perception and answer rewards during training.

## 5 CONCLUSION AND FUTURE WORK

We introduce Vision-SR1, a self-rewarded reinforcement learning framework that decomposes vision-language understanding into visual reasoning and language reasoning components. Our approach uses the VLM itself to generate explicit rewards for visual understanding, then applies Multi-Reward Policy Optimization to provide clear gradient attribution and backpropagation pathways for each reward component. Vision-SR1 strengthens visual perception and reduces language shortcuts, thereby improving VLM performance across several domains of vision-language tasks. Our proposed metric LSR further shows how perception reward lowers the tendency of models to answer via language shortcut rather than genuine visual reasoning.

This work opens up several future research directions. First, future work can focus on improving the efficiency of the *visual reasoning then think* generation format by treating the visual reasoning component as latent thinking, thereby reducing the number of decoded tokens while still enabling reward attribution to latent visual processes during the RL phase. It is also important to recognize that some of the observed mathematical gains from RL training in VLMs may come from spurious effects – for instance, recalibrating the LLM backbone's output distribution can boost multimodal math performance without true visual grounding (Shao et al., 2025). This suggests that improvements in accuracy may sometimes reflect better exploitation of language shortcuts rather than genuine perception gains. Therefore, future work can also explore more analysis to disentangle visual grounding from shortcut learning.

## ACKNOWLEDGMENTS

Boyd-Graber is supported by nsf grant 2229885, the University of Maryland (umd) TRAILS (Trustworthy ai in Law  Society) initiative. Any opinions, findings, and conclusions or recommendations

expressed in this material are those of the researchers and do not necessarily reflect the views of the National Science Foundation.

## REPRODUCIBILITY STATEMENT

To ensure the reproducibility of our research, we provide information regarding our prompt templates and experimental setup in the main paper and Appendix. All datasets and code will be released upon conference decision release.

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

# A  APPENDIX

## A.1  THE USE OF LARGE LANGUAGE MODELS (LLMS)

We acknowledge the use of large language models (LLMs) as assistive tools in this research. Our use of LLMs was limited to refine grammar and improve language clarity. All outputs from these models were meticulously reviewed, revised, and verified by the authors, who retain full responsibility for all content presented in this paper.

# B  EXPERIMENT DETAILS

## B.1  PROMPT TEMPLATES

This section presents the prompt templates used for constructing the cold start training data and Model Training prompt. The ***See-Think*** prompt is used for generating SFT ***See-Think*** data and model training. The Caption-Reasoner prompt is used to generate text-only caption reasoner SFT data and self-reward during training.

---

**_See-Think_ Prompt Template**

{*Question*}
You are tasked with analyzing an image/video to generate a detailed description to help you answer the question. First analyze the image/video and produce a self-contained description—detailed enough that can lead to the correct answer. Wrap the entire description in $< description >< /description >$ tags.

Next, engage in an internal dialogue and include self-reflection or verification in your reasoning process. Provide your detailed, step-by-step reasoning based on the image/video description information and image/video, and enclose this part within $< think >< /think >$ tags.

Finally, provide a single word or phrase answer to the question in \boxed{}.

The output format should be: $< description >$ image/video description here $< /description > < think >$reasoning process here $< /think >$ \boxed{FINAL ANSWER here}.

*Note:* {Question} *is a placeholder for the actual question.*

---

**Caption-Reasoner (Self-Reward) Prompt Template**

Text description: {Description}

Question: {Question}

You are provided a text description of a problem and a question. Determine the answer to the question based on the text description. First provide an internal step-by-step reasoning within $< think >< /think >$ tags, then provide a single word or phrase answer in \boxed{}.

*Note:* {Description} *is a placeholder for the actual text caption.* {Question} *is a placeholder for the actual question.*

---

Table 5: Through self-reward, the model is implicitly rewarded for text-only reasoning, leading to improved performance in general reasoning and reduced degradation in math reasoning benchmarks.

| Model | MMLU-Pro | SuperGPQA | GSM8K | MATH-500 |
|---|---|---|---|---|
| *Backbone model: Qwen2.5-VL-3B* | | | | |
| Before RL | 34.3 | 15.1 | 78.5 | 65.2 |
| Vision-R1 | 47.7 | 23.1 | 82.2 | 66.0 |
| Vision-SR1 | 48.1 | 23.2 | 83.2 | 68.6 |
| *Backbone model: Qwen2.5-VL-7B* | | | | |
| Before RL | 33.4 | 17.1 | 86.0 | 73.4 |
| Vision-R1 | 53.4 | 26.7 | 85.5 | 68.2 |
| Vision-SR1 | 56.1 | 26.3 | 87.6 | 70.8 |

---

**Vision Reasoner (CoT) Prompt Template**

Question: {Question}

You FIRST think about the reasoning process as an internal monologue and then provide the final answer. The reasoning process MUST BE enclosed within $< think >< /think >$ tags. The final answer MUST BE put in \boxed{}.

*Note: {Question} is a placeholder for the actual question.*

---

## B.2 LLM-AS-A-JUDGE PROMPT

We use Gemini-2.5-flash as our LLM-as-a-Judge to evaluate

---

**LLM-as-a-Judge Prompt Template**

- **Model**: Gemini-2.5-flash

**Prompt Message:**
Question: {Question}

Reference: {Reference}

Candidate: {Candidate}

You are provided a question, a gold answer, and a candidate answer. Your task is to judge the correctness of the candidate answer. Return your judgment enclosed with $< judgment >< /judgment >$.

*Note: {Question} is a placeholder for the actual question; {Reference} is a placeholder for the gold answer; {Candidate} is a placeholder for the model response.*

---

### B.2.1 ANALYSIS ON TEXT-ONLY REASONING

An interesting question is how different training strategies affect the text-only reasoning capabilities of VLMs. In particular, by decoupling visual perception and language reasoning with two separate rewards, we ask whether these abilities can mutually reinforce one another. To examine this, we evaluated the text-only performance of VLMs after RL fine-tuning on multimodal data.

Specifically, we tested on four text-only datasets: MMLU-Pro and SuperGPQA (multi-disciplinary, general-domain benchmarks), and MATH-500 and GSM8K (mathematical reasoning tasks). Our results (Table 5) compare Vision-R1, our method, and pre-RL training baselines.

First, we observe that on GSM8K and MATH-500, multimodal RL training, including both Vision-R1 and our method, degrades text-only reasoning performance. This observation aligns with recent findings on "text-only forgetting" in VLMs Zhang et al. (2024b); Ratzlaff et al. (2025), which show

Table 6: Results of ablation study: Vision-SR1 v.s. Vision-SR1 w/o visual perception self-reward.

| Methods | General Visual Understanding | | | | Visual Math & Hallucination | | | |
| | MMMU -Pro | MMMU | RealWorld QA | VisNum Bench | Math Verse | MATH -Vision | Hallusion Bench | Avg. |
|---|---|---|---|---|---|---|---|---|
| Vision-SR1 (3B) | 40.8 | 49.6 | 66.1 | 41.9 | 45.8 | 29.3 | 68.3 | 48.8 |
| ⊢ w/o self-reward | 40.0 | 48.0 | 62.6 | 41.6 | 45.1 | 30.2 | 65.8 | 47.6 |
| Vision-SR1 (7B) | **40.7** | **52.2** | **69.2** | **43.5** | **54.5** | **36.2** | **68.9** | **52.2** |
| ⊢ w/o self-reward | 42.8 | 51.8 | 67.3 | 35.7 | 52.6 | 34.4 | 67.8 | 50.3 |

that visual instruction tuning can impair language reasoning (particularly in mathematics) depending on the underlying LLM. Second, compared to Vision-R1, our method proved more effective at mitigating performance degradation on text-only mathematical benchmarks (MATH-500, GMS8K) and yielded larger gains on general knowledge tasks (MMLU-Pro, SuperGPQA). This indicates that separating the optimization signals for visual perception and language reasoning helps preserve text-only competencies, while still enabling improvements from multimodal training.

### B.2.2 ABLATION STUDY ON SELF-REWARD

We train a control version of our model without the visual reasoning self-reward and Multi-Reward Policy Optimization (Vision-SR1 w/o self-reward). This ablated model still follows a structured output (visual perception, CoT reasoning, and answer) but is optimized only with answer and format rewards. The self-visual reward for self-evaluating visual reasoning and Multi-Reward Policy Optimization are removed. We note the only difference between Vision-SR1 w/o self-reward and Vision-R1 (Huang et al., 2025b) lies in the output structure, i.e., using different system prompts, while all supervision signals (answer reward and format rewards) remain the same. Interestingly, our system prompt yields slightly better performance (+1.0 on average). Table 6 reports the ablation results. We find that not including visual reasoning reward and Multi-Reward Policy Optimization could lead to overall worse VLM task performance compared to including them in the training process.

