# OpenReview forum: "Vision-SR1: Self-Rewarding Vision-Language Model via Reasoning Decomposition and Multi-Reward Policy Optimization"
_ICLR.cc/2026/Conference — ICLR 2026 Poster_

### Official Review · Reviewer_mWSw · 2025-10-18

**Soundness:** 3
**Presentation:** 3
**Contribution:** 3
**Rating:** 6
**Confidence:** 3

**Summary:**

The paper proposes Vision-SR1, which decomposes VLM reasoning into visual perception and language reasoning, then self-verifies whether the perception alone is sufficient to answer the question, providing a self-visual reward without external supervision to reduce hallucinations and language shortcuts.

**Strengths:**

- Clear and well-written.
- Method is effective across many tasks with thorough experimental validation.
- Provides intuitive theoretical analysis supporting the design choices.
- Raises several valuable, thought-provoking questions for the community.

**Weaknesses:**

- Visual perceptions that are sufficient to answer the question without referring back the input image: is it possible (some visual information is hard to express in language)? Is it necessary (visual reasoning might be understood as a carrier of latent space reasoning, used to aggregate and evolve visual information and implicit thinking via attention)?
- “If the correct answer is derived, a self-visual reward is assigned.” Which part of generation is this visual reward used to encourage in the paper? I am a bit confused. Can this visual reward be assigned solely to the visual perception part in the first rollout? What would the effect be?
- Only Qwen-2.5-VL is trained; how about other model series? Adding them would be more convincing.
- Is the Language Shortcut Rate metric reasonable? First, “If the evaluator can reproduce the correct ground-truth answer using only this information, the generated visual reasoning is deemed self-contained.” How many cases are there where the evaluator reproduces the wrong answer but the generated visual reasoning is actually self-contained? It would be better to have a more detailed human analysis (on a small amount of data is fine).
- Curious about the method’s performance on spatial reasoning tasks, because I think many perceptions and reasoning in 3D space are hard to express in natural language.
- Typo: an extra comma at line 067.

**Questions:**

- Additional discussion: Whether the generated visual reasoning contains all information needed to answer the question—Is it possible, and is it necessary?

- Additional experiments:
  - If possible, try assigning the visual reward solely to the visual perception part in the first rollout.
  - If time permits, train models beyond the Qwen-2.5-VL series.
  - If time permits, evaluate Vision-SR1 on spatial reasoning tasks such as VSI-Bench [1], MMSI-Bench [2], and OmniSpatial [3].

- Additional human analysis: In the Language Shortcut Rate evaluation, how many cases are there where the evaluator reproduces the wrong answer while the generated visual reasoning is actually self-contained? It would be better to include a more detailed human analysis (a small amount of data is sufficient).


[1] Yang, Jihan, et al. "Thinking in space: How multimodal large language models see, remember, and recall spaces." Proceedings of the Computer Vision and Pattern Recognition Conference. 2025.

[2] Yang, Sihan, et al. "MMSI-Bench: A Benchmark for Multi-Image Spatial Intelligence." arXiv preprint arXiv:2505.23764 (2025).

[3] Jia, Mengdi, et al. "OmniSpatial: Towards Comprehensive Spatial Reasoning Benchmark for Vision Language Models." arXiv preprint arXiv:2506.03135 (2025).

---

> ### Author Response · Authors · 2025-11-22
> **Response to weakness 1**
>
> Is it possible (some visual information is hard to express in language)
> - Yes. It is possible that not all visual information can be expressed in language. Some visual cues (fine textures, subtle spatial relations, color gradients, dense numbers, etc.) are hard to verbalize. If they cannot be faithfully captured in natural language, a second-pass answer that sees only the caption might still fail—even if the first-pass caption is “as good as it gets. A recent work shows that instead of using explicit text description, expressing the visual information in json format to show relative locations between objects can also improve visual ability with two-stage reward process [1].
>
> Visual reasoning might be understood as a carrier of latent space reasoning
> - Language acts as a low-variance supervision channel. When the model must generate the evidence it relies on, reward aligns with verifiable, human-checkable facts instead of opaque hidden states. When only using latent visual to solve multimodal questions, it becomes hard to verify the correctness of the visual information thus hard to provide verifiable rewards. This is the same motivation behind Concept/Language Bottleneck Models — enforcing an interpretable intermediate representation improves both robustness and generalization [2].
>
> We provide a quantitative example showing where latent space visual reasoning is less effective and interpretable than the direct language description of multimodal questions.:
>
> ### Example question:
> ************************************************************************************
> \<image\> Please answer the question embedded in the image.
> ************************************************************************************
>
> ### Model Caption:
> The image shows a problem about calculating the volume of water in a reservoir formed by a dam across a valley. The table lists contour elevations (in meters) and the corresponding areas (in square meters) enclosed by those contours. The contours and their respective areas are as follows:
>
> • Contour 660 m: Area = 5200 m²
> • Contour 665 m: Area = 9400 m²
> • Contour 670 m: Area = 16300 m²
> • Contour 675 m: Area = 22400 m²
> • Contour 680 m: Area = 40700 m²
> • Contour 685 m: Area = 61500 m²
> • Contour 690 m: Area = 112200 m²
> • Contour 695 m: Area = 198100 m²
> • Contour 670 m: Area = 272400 m²
>
> The problem asks for the volume of water when the reservoir level reaches 700 m, using the prismoidal formula. The options for the calculated volume are:
>
> (A) 29.193 × 10⁶ m³
> (B) 28.893 × 10⁶ m³
> (C) 28.493 × 10⁶ m³
> (D) 29.293 × 10⁶ m³
> (E) 30.393 × 10⁶ m³
> (F) 30.093 × 10⁶ m³
> (G) 29.493 × 10⁶ m³
> (H) 29.893 × 10⁶ m³
> (I) 29.793 × 10⁶ m³
> (J) 29.693 × 10⁶ m³
> ************************************************************************************
>
> In this case, the model is extracting the information out from the image that contains the answer choices rather than giving a final answer. Especially when the questions are embedded in the visual contents, describing them in language before reasoning becomes more important.
>
> [1]   IR3D-Bench: Evaluating Vision-Language Model Scene Understanding as Agentic Inverse Rendering https://arxiv.org/abs/2506.23329
>
> [2] Language Guided Concept Bottleneck Models for Interpretable Continual Learning https://openaccess.thecvf.com/content/CVPR2025/papers/Yu_Language_Guided_Concept_Bottleneck_Models_for_Interpretable_Continual_Learning_CVPR_2025_paper.pdf

---

> ### Author Response · Authors · 2025-11-22
> **Response to clarification of visual reward**
>
> ### Which part of generation is this visual reward used to encourage in the paper?
>
> The visual reward is not applied to the perception span in the first rollout; instead it trains the answer/COT tokens generated in the second rollout. The intuition (stated in the paper 244-251) is that if the perception is self-contained the model should also be able to articulate a correct answer from it; therefore the reward teaches the model to reason to rely on faithful perceptions.
>
>
> ### Can this visual reward be assigned solely to the visual perception part in the first rollout? What would the effect be?
>
> If the visual reward is assigned solely to the visual perception part in the first rollout, it would increase the chance of reward hacking (embedding the final answer in caption because we are directly optimizing loss to the visual perception). Here is an explanation:
>
> - During optimization, the log-probabilities of visual perception tokens are multiplied by zero when computing the self-visual advantage.
> No gradient ⇒ no direct incentive to hack the answer. Instead, the visual reward that is broadcast (to second stage COT and answer tokens) can flow through shared parameters, but it changes the visual perception token distribution only indirectly and weakly—much less than if the visual perception itself carries the advantage.
>
> - In our proposed method, the visual perception tokens generated in the first stage are treated as context rather than actions in the second rollout. This design encourages the VLM to identify and extract important visual details from images before engaging in reasoning. In other words, the visual component provides contextual grounding for reasoning, rather than serving as a direct optimization target that leads the model to embed the final answer into the visual perception before reasoning.

---

> ### Author Response · Authors · 2025-11-22
> **Response to training on other families of models**
>
> We extend our training to other families of models other than Qwen-VL series: Mimo-7B-SFT
>
> We provide the results of Mimo below:
>
> | Model                  | MMMU-Pro | MMMU | ReadWorldQA | VisNumBench | MathVerse | Math-Vision | Hallusionbench | Avg  |
> | ---------------------- | -------- | ---- | ----------- | ----------- | --------- | ----------- | -------------- | ---- |
> | Mimo-7B-Base           | 38.0     | 45.6 | 68.2        | 30.2        | 35.5      | 21.6        | 71.9           | 44.4 |
> | Mimo (Vision-R1)       | 38.7     | 47.3 | 67.1        | 33.5        | 35.3      | 25.7        | 74.3           | 46.0 |
> | Mimo (Vision-SR1 ours) | 39.3     | 49.5 | 68.1        | 44.6        | 40.0      | 29.6        | 75.6           | 49.5 |
>
> The trend also generalizes to models other than Qwen-VL.
>
> [2] Mimo-VL Technical Report: https://arxiv.org/abs/2506.03569

---

> ### Author Response · Authors · 2025-11-22
> **Response to spatial dataset understanding**
>
> We extend our testing benchmarks to three additional datasets that test spatial understanding and a language-shortcut evaluation dataset mentioned by reviewer [BBYi](https://openreview.net/forum?id=C1M4ETatgM&noteId=yVrJmGPKyb).
>
>
> We show the results of Qwen2.5-VL-7B and Mimo-7B on these language short-cut and spatial intelligence datasets.
>
> | Model                  | ViLP (language priors) | MMSI-Bench | OmniSpatial | Avg  |
> | ---------------------- | ---------------------- | ---------- | ----------- | ---- |
> | Mimo-7B-Base           | 56.4                   | 28.2       | 40.3        | 41.6 |
> | Mimo (Vision-R1)       | 58.2                   | 27.7       | 40.4        | 42.1 |
> | Mimo (Vision-SR1 ours) | 59.3                   | 28.0       | 42.7        | 43.3 |
> | Qwen-7B-Base           | 45.1                   | 24.0       | 27.3        | 32.1 |
> | Qwen (Vision-R1)       | 51.3                   | 21.9       | 31.1        | 34.8 |
> | Qwen (Vision-SR1 ours) | 52.6                   | 27.7       | 44.2        | 41.5 |
>
> Scene understanding is highly vision-intensive and therefore less susceptible to language priors that might otherwise allow the model to guess the correct answer without truly perceiving the visual content.
>
>
> [2] Mimo-VL Technical Report: https://arxiv.org/abs/2506.03569
>
> [3] ViLP: Probing Visual Language Priors in VLMs and Winoground https://arxiv.org/pdf/2501.00569
>
> [4] MMSI-Bench: A Benchmark for Multi-Image Spatial Intelligence https://arxiv.org/abs/2505.23764
>
> [5] OmniSpatial: Towards Comprehensive Spatial Reasoning Benchmark for Vision Language Models https://arxiv.org/abs/2506.03135

---

> > ### Comment · Reviewer_mWSw · 2025-11-22
> >
> > Thank you for your detailed and thoughtful response. However, I feel that my earlier concern:
> >
> > >“Additional human analysis: In the Language Shortcut Rate evaluation, how many cases are there where the evaluator reproduces the wrong answer while the generated visual reasoning is actually self-contained? It would be better to include a more detailed human analysis (a small amount of data is sufficient).”
> >
> > has not been fully addressed.
> >
> > I would still encourage the authors to conduct and report a brief but more explicit human analysis along these lines, even on a small subset of data.
> >
> > Overall, I view the paper as marginally above the acceptance threshold. I am therefore keeping my rating unchanged, but I have increased my confidence in this assessment.

---

> > > ### Author Response · Authors · 2025-11-22
> > > **Response to human analysis**
> > >
> > > Thank you for your feedback. We have not yet posted the human-analysis results because we needed additional time to thoroughly review the dataset and model outputs; the manual analysis process took longer than we expected.
> > >
> > > - ### Is the Language Shortcut Rate metric reasonable? First, “If the evaluator can reproduce the correct ground-truth answer using only this information, the generated visual reasoning is deemed self-contained.”
> > >
> > > Language-Shortcut Rate (LSR) measures how often a model answers correctly only because it sees the image, not because its self-generated caption is informative. We compute LSR by (i) hiding the image, (ii) giving a stronger LLM (Gemini-2.5-pro) just the caption + question, and (iii) counting cases where this LLM now fails although the full vision model has succeeded. For this LSR evaluation, we use a strong LLM because we want the visual caption judgment to be as accurate as possible.
> > >
> > > A high LSR therefore signals reliance on language priors instead of visual grounding. While a few images may be intrinsically hard to verbalize, our benchmarks feature mostly 2-D scenes with easily describable attributes, so LSR can be a reliable indicator of language shortcut behaviors.
> > >
> > > - ### How many cases are there where the evaluator reproduces the wrong answer but the generated visual reasoning is actually self-contained?
> > >
> > > We compute the percentage of answers where VLM produces wrong final answer but the Gemini-2.5-Pro judges the visual captioning as correct below:
> > >
> > > |   Model  | MMMU-Pro | MMMU | ReadWorldQA | VisNumBench | MathVerse | Math-Vision | Hallusionbench | Avg |
> > > |-----------|-----------|------|--------------|--------------|-----------|--------------|----------------|------|
> > > | 7B-Vision-SR1 | 11.1 | 7.3 | 4.8 | 14.3 | 10.9 | 14.1 | 5.1 | 9.7 |
> > >
> > > These percentages mean the VLMs have limited reasoning abilities to solve the question even though the given visual captions are informative enough. This means a powerful and strong enough reasoning model can solve the question by giving enough visual caption information. But due to the VLM’s current limited reasoning ability, even giving it an informative visual caption and image, it cannot solve the problem.

---

> > > > ### Comment · Reviewer_mWSw · 2025-11-23
> > > >
> > > > Thank you for your detailed analysis and response.

---

> ### Author Response · Authors · 2025-11-22
> **Small Human Subset Analysis #1**
>
> We carefully select twenty representative samples from the poll (Vision-SR1 has the wrong final answer but visual caption is judged correct by Gemini) and analyzed whether these visual captions judged by Gemini-2.5-Pro actually align with human judgments. That is, given the visual captions generated from Vision-SR1 alone and the question, whether it is possible to answer the question. Out of all the visual captions we analyzed, they indeed correlate with human judgments. Thus, these are examples that shows although the VLM can recognize all the visual information to solve the problems, but due to its size and LLM backbone reasoning abilities, it is unable to solve and get the final correct answer. We show two of the example questions and visual captions below (We can provide more examples if you wish to see more):
>
>
>
> ### Question 1:
>
> In the Gluteal region and thigh, identify the 157 structure. \nA. Superior gluteal nerve\nB. Gluteus minimus\nC. Semitendinosus\nD. Piriformis.
>
> —---------------------------------------------------------------------
>
> ### Gold Answer: D
>
> —---------------------------------------------------------------------
>
> ### Full VLM Output (With image input):
>
> \<description\>
> The diagram shows a posterior view of the gluteal region and thigh. The gluteus maximus forms the broad, outermost layer of the buttocks. Superficially, the gluteus maximus is labeled with a number (likely 150). The 157 structure is located superiorly in the gluteal region, appearing as a small, lateral, and somewhat triangular muscle that is positioned just deep to the gluteus maximus. It is the piriformis muscle, a short, horizontally oriented muscle that is part of the lateral rotator group and lies deep to the gluteus maximus. The other labeled structures in the vicinity include the superior gluteal artery and nerve (often found superiorly), and the gluteus medius and minimus muscles, which are more laterally placed. The semitendinosus is a long, posterior thigh muscle, not typically found in the gluteal region./n\</description\>
>
> \<think\>
> To identify the structure labeled as 157 in the gluteal region and thigh, I need to examine the anatomical diagram provided. The image shows various muscles and nerves in the gluteal region and thigh area.
>
> First, I locate the area labeled 157. It is situated in the gluteal region, which is the area of the buttocks. The structure labeled 157 appears to be a small muscle located superiorly in the gluteal region.
>
> Next, I consider the options provided:
> A. Superior gluteal nerve - This is more likely to be a nerve rather than a muscle, and it would be positioned superiorly in the gluteal region.
> B. Gluteus minimus - This is a muscle in the gluteal region, and its position matches the location of 157.
> C. Semitendinosus - This muscle is located more medially in the thigh, not in the gluteal region.
> D. Piriformis - This muscle is also in the gluteal region and is known to be in this area. However, the piriformis is typically more medially located compared to the location of 157.
>
> Given the location and the size, the structure labeled 157 is consistent with the gluteus minimus, one of the muscles in the gluteal region.
> \</think\>
>
> \boxed{B} --> Wrong answer
>
> —---------------------------------------------------------------------
>
> ### Gemini visual caption Verify Responses (Using just visual caption and question without image):
> The question asks to identify structure 157 in the gluteal region and thigh.
> The text describes structure 157 as located superiorly in the gluteal region, appearing as a small, lateral, and somewhat triangular muscle that is positioned just deep to the gluteus maximus. It is identified as the piriformis muscle.
>
> Therefore, the answer is Piriformis.
>
> A. Superior gluteal nerve - The text mentions the superior gluteal artery and nerve, but does not identify it as 157.
> B. Gluteus minimus - The text mentions the gluteus minimus, but does not identify it as 157.
> C. Semitendinosus - The text mentions the semitendinosus, but does not identify it as 157.
> D. Piriformis - The text identifies the 157 structure as the piriformis muscle.
>
> \boxed{D. Piriformis} -> The visual caption is informative.

---

> > ### Author Response · Authors · 2025-11-22
> > **Human analysis example 2**
> >
> > ### Question:
> > Skin tests were used to screen for hepatic schistosomiasis in rural villages along the river and the results were shown in the table. The Pr+ was\nA. 0.69\nB. 0.98\nC. 0.02\nD. 0.94
> >
> > ---------------------------------------------------------------------
> >
> > ### Gold Answer: A
> >
> > ---------------------------------------------------------------------
> >
> > ### Model Responses:
> > \<description\>
> > A table is shown with two rows (positive and negative) and two columns (hepatic schistosomiasis: Yes and No). The totals are:
> > • Positive test results: 117 (Yes) + 53 (No) = 170
> > • Negative test results: 8 (Yes) + 312 (No) = 320
> > • Total individuals: 125 (Yes) + 365 (No) = 490
> >
> > The question asks for the positive predictive value (Pr+) of the test, which is calculated as (Number of true positives) / (Number of true positives + Number of false negatives). Here, true positives are the 117 individuals who tested positive and have hepatic schistosomiasis, and the number of false negatives is the 8 individuals who have hepatic schistosomiasis but tested negative. Therefore, Pr+ = 117 / (117 + 8) = 117 / 125 = 0.936, which rounds to 0.94./n\</description\>
> >
> > \<think\>
> > To find the positive predictive value (Pr+) of the test, we need to use the formula:
> >
> > \[ \text{Pr+} = \frac{\text{True Positives}}{\text{True Positives} + \text{False Negatives}} \]
> >
> > From the table:
> > - True Positives (TP) are the number of people who tested positive and have hepatic schistosomiasis. According to the table, this number is 117.
> > - False Negatives (FN) are the number of people who have hepatic schistosomiasis but tested negative. According to the table, this number is 8.
> >
> > Now, we can plug these values into the formula:
> >
> > \[ \text{Pr+} = \frac{117}{117 + 8} = \frac{117}{125} \]
> >
> > Calculating the fraction:
> >
> > \[ \frac{117}{125} = 0.936 \]
> >
> > Now, we compare this value to the options given:
> > A. 0.69
> > ...
> > Therefore, the positive predictive value (Pr+) is 0.94.
> > \</think\>
> >
> > \boxed{D}
> >
> > —---------------------------------------------------------------------
> >
> > ### Gemini Visual Caption Eval Responses:
> > The text description clearly explains the calculation of the positive predictive value (Pr+).
> > The formula is given as (Number of true positives) / (Number of true positives + Number of false positives).
> > The text identifies the number of true positives as 117 and the number of false positives as 53.
> > Therefore, Pr+ = 117 / (117 + 53) = 117 / 170 = 0.688 which rounds to 0.69.
> >
> > The text description incorrectly calculates the positive predictive value (Pr+). The correct formula is (Number of true positives) / (Number of true positives + Number of false positives). The text incorrectly uses the number of false negatives in the denominator.
> >
> > The correct calculation is:
> > True Positives = 117
> > False Positives = 53
> > Pr+ = 117 / (117 + 53) = 117 / 170 = 0.688235... which rounds to 0.69
> >
> > \</think\>
> > \boxed{A. 0.69}
> >
> > —---------------------------------------------------------------------
> >
> > This example also shows the problem requires complex math reasoning and computation, although the VLM is able to extract out all the number information from the image accurately, it fails to solve this problem. But given a stronger model such as Gemini-2.5-Pro, it is able to solve the problem using the extracted visual information from the VLM.

---

### Official Review · Reviewer_ExG7 · 2025-10-31

**Soundness:** 3
**Presentation:** 3
**Contribution:** 4
**Rating:** 6
**Confidence:** 4

**Summary:**

This paper proposes Vision-SR1, a self-reward reinforcement learning (Self-RL) framework designed to address visual hallucinations and language shortcuts in VLMs. It decomposes reasoning into two stages: visual perception and language reasoning. The model first generates a self-contained perceptual description (c), then re-prompts itself and relies solely on c to answer the question, thereby generating a self-derived visual reward. The method employs a multi-reward strategy for optimization, computing decoupled losses for both visual and answer rewards, effectively enhancing visual reasoning capability while reducing the LSR.

**Strengths:**

(1) Vision-SR1 allows VLMs to self-verify whether their visual perception is self-contained and sufficient to answer a given question. This effectively provides dense and adaptive reward signals for intermediate visual reasoning steps, avoiding the high costs, biases, and reward hacking risks associated with external human annotations or large model distillation.
(2) Advantage functions and loss terms are computed separately for visual and answer rewards, effectively decoupling the training signals. This ensures balanced reinforcement of both visual perception and language reasoning, while also avoiding the signal sparsity and entanglement caused by the traditional approach of summing rewards.

**Weaknesses:**

(1) Vision-SR1 uses the model itself as a verifier, evaluating the quality of visual perception by re-prompting the same-policy VLM. This may lead to circular dependencies and inherent biases: the model could learn to generate perceptions that "convince itself" rather than perceptions that truly align with the image.
(2) By re-prompting the same-policy VLM to assess visual perception quality, the method relies on two rollouts during training, which may effectively double the training cost.

**Questions:**

The authors mention that self-rewarding may lead to reward hacking. Could the authors quantitatively analyze the intrinsic bias of the reward model in Vision-SR1? Furthermore, would it be possible to introduce lightweight external verification (e.g., a frozen CLIP or BLIP module) to cross-check the true visual consistency of the self-reward, thereby mitigating the circular dependency issue inherent in self-rewarding?

---

> ### Author Response · Authors · 2025-11-22
> **Response to Possible Reward Hacking**
>
> In the first rollout, the model generates a three part output <caption><CoT><answer>, where the second rollout indirectly validates the correctness of caption based on the ground truth answer the model generates from the caption. Instead of directly evaluating the caption part, indirect evaluation does not and less-likely lead to direct reward hacking [1].
>
> We provide analysis why our two-stage rewarding can be more robust to reward hacking:
>
> - During optimization, the log-probabilities of visual perception tokens are multiplied by zero when computing the self-visual advantage.
> No gradient ⇒ no direct incentive to hack the answer. Instead, the visual reward that is broadcast (to second stage COT and answer tokens) can flow through shared parameters, but it changes the visual perception token distribution only indirectly and weakly—much less than if the visual perception itself carries the advantage.
>
> - In our proposed method, the visual perception tokens generated in the first stage are treated as context rather than actions in the second rollout. This design encourages the VLM to identify and extract important visual details from images before engaging in reasoning. In other words, the visual component provides contextual grounding for reasoning, rather than serving as a direct optimization target that leads the model to embed the final answer into the visual perception before reasoning.
>
> In addition, using external evaluators could also lead to extrinsic biases and reward hacking. Resolving Reward hacking is out of scope for this paper and we will provide quantitative analysis below.
>
>
> [1] Internal Consistency and Self-Feedback in Large Language Models: A Survey https://arxiv.org/abs/2407.14507

---

> ### Author Response · Authors · 2025-11-22
> **Response to Compute Efficiency**
>
> Thank you for your critical review and similar concerns as reviewer f7j6.
>
> Our proposed method does not double the training cost compared to regular GRPO training. Our training approach is an end-to-end approach that involves any external models to assign visual rewards for VLMs. Right after the model generates the first rollout, we take the output from the first rollout and use the same model to perform generation for the second rollout using VLLM, which speeds up batch inferencing for the second rollout without loading additional models.
>
> 1. Compared to the standard one stage GRPO approach, because we are using the same VLM to perform the second stage rollout without loading external models or interacting with API models, the training time is only 10%-15% longer than the one stage training, not exactly twice longer. We reviewed the training logs between one step GRPO and two-stage self-reward GRPO. We see that training for 20 steps (per device batch size 8 on 8 GPUs) takes regular GRPO around 10.5 hours, and takes two stage training around 12 hours.
>
> 2. We also experimented using other external models for the two-stage training to judge the visual caption correctness generated by the model. We used GPT-4o as an external model and we quickly reached the frequency and rate limit of the API call since for each data point there are 8 rollout responses and per GPU we have batch size 8 * 8=56 responses to let the API evaluate. We not only hit the rate limit quickly, but the training time also increased more than doubled– 20 steps for around more than twenty hours.
>
> If we use an external open-sourced local model, we also need to spare a GPU on our node to deploy it. If we have 8 GPUs on our node, to train the model with an external judge, we need to spare one GPU for it to run inference. That means we only have seven GPUs left for model training, which we need to count for the external model delay and fewer GPUs for training delay. Another contribution is that we leverage the self-reward process to make the two-stage training more efficient than relying on external models while not doubling the training costs.

---

> ### Author Response · Authors · 2025-11-23
> **Response to Intrinsic bias of visual reward analysis**
>
> We think the intrinsic bias you are referring to is a bias analysis to quantify the visual perception caption faithfulness. We introduced a Language Shortcut Rate (LSR) that measures the reward model biases (Section 3.3.3). In the LSR rate, we use an external more powerful LLM (Gemini-2.5-Pro) to judge whether the visual captions are self-contained. In this evaluation, since we are using an external LLM to evaluate, we also implicitly evaluated the model’s intrinsic biases.
>
> The LSR measures cases where visual perception is incorrect, but the final answer is still correct. This metric appears in Table 4 of the document. This is a quantitative measurement of intrinsic reward bias, because it measures how often the model rewards itself (the final answer is correct but visual caption is incorrect) even when perception is wrong.
>
> Thus, this metric also measures intrinsic bias where:
> - High LSR = Strong bias toward language priors -> reward hacking risk
> - Low LSR = Reward follows perception → less intrinsic bias.
>
> This is relevant to reward hacking analysis mentioned in the question. We also provide more insights for Table 4, and we will include them as additional analysis in our revision.
>
> - In Table 4, less language shortcut occur on visual intensive datasets -> (MMMU-Pro, MMMU 6-8 percent) → Reward strongly favors visual grounding tasks.
>
> - There is higher language shortcut on math datasets compared to the visual intensive datasets (MathVerse and Mathvision 8-10 percent) → The visual reward model is less helpful and reasoning reward (final answer) takes over.
>
> The result indicates that the self-reward has a bias toward tasks requiring explicit visual grounding. It is weaker for symbolic math where language reasoning dominates. This is what we want for the training since we want the VLM to focus more on visual when the question if visual intensive and focus more on reasoning when the question is reasoning heavy.

---

> > ### Author Response · Authors · 2025-11-23
> > **Response to external verification**
> >
> > We agree that external verification is a promising direction to further validate the visual consistency of VLM’s visual perceptions. To examine whether the self-reward signal suffers from circular dependency, we conducted an additional analysis using a frozen CLIP model to independently evaluate the similarity between the input image and the visual perceptions produced by Vision-SR1.
> > We computed the image–caption CLIP similarity scores between the input image and the model-generated visual reasonings. The averaged CLIP scores across tasks are summarized below:
> >
> >
> > | Model  | MMMU-Pro | MMMU | RealWorldQA | VisNumBench | MathVerse | Math-Vision | Hallusionbench | Avg  |
> > |--------|----------|------|-------------|-------------|-----------|-------------|----------------|------|
> > | Vision-SR1-7B | 0.65  | 0.66 | 0.63  | 0.67  | 0.64 | 0.65  | 0.67  | 0.65 |
> >
> > - The results in the table show that CLIP similarity is high and stable across datasets. The CLIP scores remain consistently high (0.63–0.67) across all seven benchmarks. This indicates that the visual perceptions produced by our model are stably aligned with the input image—not merely self-consistent with the model’s own language priors.
> >
> >
> > - Visual–text inconsistency likelihood is low If circular dependency is present, Vision-SR1's visual descriptions would become increasingly text-biased and reasoning-heavy, particularly on tasks like MathVision and MathVerse where correct answers can often be inferred through linguistic patterns without accurate visual grounding. This will likely lower CLIP image-caption alignment on perception-intensive benchmarks (e.g., MMMU-Pro, VisNumBench) where visual perception is important. In our results, we see that perception-heavy tasks achieve consistent and similar CLIP scores (0.65–0.67) as reasoning heavy datasets (mathvision and mathverse),  showing that the self-reward mechanism does not degrade visual grounding quality and only incentivize text-only reasoning.
> >
> >
> >
> > - The stable CLIP–caption scores across various datasets shows that a frozen, external model independently confirms the visual consistency of the generated captions. This shows that Vision-SR1’s self-reward is not merely “self-convincing,” but aligns with an external vision encoder’s evaluation.
> >
> > We will add these findings in our revision.

---

### Official Review · Reviewer_f7j6 · 2025-11-01

**Soundness:** 3
**Presentation:** 3
**Contribution:** 3
**Rating:** 4
**Confidence:** 4

**Summary:**

The authors propose an improvement to vision-R1, decoupling the visual perception description generation part and the answer generation part for reward, to avoid LVLMs overly relying on language reasoning shortcuts. The method is clear, the experiments are solid, and the comparison with vision-R1 is clear.

**Strengths:**

1. The writing is clear, the method description is clear and specific, very straightforward, and the experiments are solid.

**Weaknesses:**

1. The only concern is novelty, as the idea of first generating image description and then generating answer based on the description has been explored before. [1]
2. Requires inference twice, which would be time-consuming and not infrastructure-friendly.

[1] Multimodal Chain-of-Thought Reasoning in Language Models

**Questions:**

1. Have the authors analyzed how the model's attention changes after this training, and whether it affects its distribution on visual tokens?

2. How do the authors ensure that the generated image description is self-contained? Since the model can see both the question and the image, it could potentially guess the answer and then generate an image description where the description already contains answer-related information.

3. How do the authors guarantee that decoupling the visual perception stage and the answer generation stage can avoid language shortcuts? From a training perspective, in the second stage, after removing the image input, wouldn't the model be more prone to relying on language shortcuts? So why does using this two-stage decoupling method mitigate language shortcuts and visual hallucination? Why can't a single-stage approach achieve this? Is there a theoretical explanation? For example, changes in attention patterns over visual tokens?

4. How do the authors address the infrastructure inefficiency issue for this two-stage training approach?

---

> ### Author Response · Authors · 2025-11-21
> **Response to Weakness 1: Comparison to previous works**
>
> We acknowledge that paper [1] is the first proposing generating caption then generating chain-of-thought, and we have cited it in our paper. We distinguish our work from previous works by:
>
>
> - Previous works explored either two-stage reasoning (description + CoT) using prompt engineering and SFT, or one-stage (direct CoT) RL training without explicit visual reward. The novelty of our work is to perform two-stage RL training (using GRPO) and carefully designed rewards to improve OCR and visual recognition of VLMs.
>
>
> - We propose a self-reward process, which leverages the model itself to judge its own generated visual performance, which has not been explored before. This approach opens up a new direction for using LLM-as-a-judge for multiple rewards (chain-of-thought steps [3]) scalable and possible. Previous works mainly use rule-based approaches to assign rewards during the RL training process due to the latency and cost issues of involving an LLM in the training process [4]. Our self-reward approach can make involving LLM judge in the training process possible (See [Response link Regarding Efficiency](https://openreview.net/forum?id=C1M4ETatgM&noteId=Wxp7lkR8fT) for compute efficiency details).
>
>
> [1] Multimodal Chain-of-Thought Reasoning in Language Models: https://arxiv.org/abs/2302.00923
>
> [3] R1-VL: Learning to Reason with Multimodal Large Language Models via Step-wise Group Relative Policy Optimization: https://arxiv.org/pdf/2503.12937
>
> [4] [Deepscaler-R1](https://pretty-radio-b75.notion.site/DeepScaleR-Surpassing-O1-Preview-with-a-1-5B-Model-by-Scaling-RL-19681902c1468005bed8ca303013a4e2)

---

> ### Author Response · Authors · 2025-11-21
> **Response to Weakness 2 and Question 4: Regarding efficiency**
>
> Our training approach is an end-to-end approach that involves any external models to assign visual rewards for VLMs. Right after the model generates the first rollout, we take the output from the first rollout and use the same model to perform generation for the second rollout using VLLM, which speeds up batch inferencing for the second rollout without loading additional models.
>
> 1. Compared to the standard one stage GRPO approach, because we are using the same VLM to perform the second stage rollout without loading external models or interacting with API models, the training time is only 10%-15% longer than the one stage training, not exactly twice longer. We reviewed the training logs between one step GRPO and two-stage self-reward GRPO. We see that training for 20 steps (per device batch size 8 on 8 GPUs) takes regular GRPO around 10.5 hours, and takes two stage training around 12 hours.
>
> 2. We also experimented using other external models for the two-stage training to judge the visual caption correctness generated by the model. We used GPT-4o as an external model and we quickly reached the frequency and rate limit of the API call since for each data point there are 8 rollout responses and per GPU we have batch size 8 * 8=56 responses to let the API evaluate. We not only hit the rate limit quickly, but the training time also increased more than doubled– 20 steps for around more than twenty hours.
>
> If we use an external open-sourced local model, we also need to spare a GPU on our node to deploy it. If we have 8 GPUs on our node, to train the model with an external judge, we need to spare one GPU for it to run inference. That means we only have seven GPUs left for model training, which we need to count for the external model delay and fewer GPUs for training delay. Another contribution is that we leverage the self-reward process to make the two-stage training more efficient than relying on external models while not doubling the training costs.

---

> ### Author Response · Authors · 2025-11-21
> **Response to attention map analysis**
>
> Based on the suggestions from reviewer [BBYi](https://openreview.net/forum?id=C1M4ETatgM&noteId=yVrJmGPKyb), we sample 50 images from the ViLP dataset [3] and examine the visual attention changes for the trained Qwen-VL-SR1 model and the original base model. The ViLP dataset specifically tests for language short-cut priors in VLMs.
>
> Qwen-VL-7B uses ViT as its visual encoder. To analyze how post-training affects visual attention, we compute the weight difference between the post-trained model and the pretrained model for the same set of images, and then average these differences. We then measure the percentage increase in visual attention across all layers. Specifically, we report the following layer-wise changes:
>
> **Early Feature Extraction (Layers 0 – 7)**
>
> | Layer    | Change             | Interpretation                                                                               |
> | -------- | ------------------ | -------------------------------------------------------------------------------------------- |
> | 6        | +10.16 %           | Strongest increase in entire network – model learned to extract richer visual features early |
> | 5        | +5.12 %            | Visual feature extraction                                                         |
> | 0 – 4, 7 | +0.83 % to +2.88 % | Modest improvements in initial visual processing                                             |
>
>
>
>
> **Later Layers (Layers 14 – 27)**
>
> | Layer      | Change             | Interpretation                                                   |
> | ---------- | ------------------ | ---------------------------------------------------------------- |
> | 20         | +9.15 %            | Second-strongest increase – improved late-stage visual reasoning |
> | 27         | +6.11 %            | Late layer showing strong visual grounding                      |
> | 18         | +5.97 %            | Late-stage visual attention re-engagement                                  |
>
>
>
> **Top 5 Most-Changed Layers**
>
> | Layer | L2 Distance | Visual Attention Change |
> | ----- | ----------- | ----------------------- |
> | 25    | 61.19       | +3.24 %                 |
> | 26    | 52.63       | +4.80 %                 |
> | 24    | 52.51       | +4.46 %                 |
> | 22    | 51.82       | +3.76 %                 |
> | 5     | 51.69       | +5.12 %                 |
>
>
> From the top 5 most changed layers, we see that the visual attention changed the most in the very early stage and late stage in the ViT. The models learns to extract more visual features in the beginning and late layers now better integrate visual and linguistic information before processing jointly with the LLM backbone.
>
>
> [3] ViLP: Probing Visual Language Priors in VLMs and Winoground https://arxiv.org/pdf/2501.00569

---

> ### Author Response · Authors · 2025-11-21
> **Response to visual captions are self-contained**
>
> In our training prompt, we specifically asked the VLM to generate the visual description for objects, relevant information such as OCR or number extraction without generating the final answer in it so the caption only contains necessary and details from the images.
>
>
> Additionally, we extract all the description from our model outputs from the benchmarks and use LLM-as-a-judge (Qwen-2.5-14B-Instruct) and regex matching to judge whether the visual description contains the final answer. Among the benchmarks we tested in the paper across around 1,1000 multimodal QA pairs, 32% of the visual captions contain the final answers. However, among those descriptions that contain the final answers, models do not necessarily output the final correct answer for all of them. Rather, the model is describing the answers from the questions. For example, in MMMU-Pro-Vision subset, where there is no text input as questions. All the questions and choices are embedded in the image. In that case, the model is extracting the questions and choices out as the description instead of generating the final answer in the description part.
>
> ### Example question:
>
> \<image\> Please answer the question embedded in the image.
> ************************************************************************************
>
> ### Model Caption:
>
> The image shows a problem about calculating the volume of water in a reservoir formed by a dam across a valley. The table lists contour elevations (in meters) and the corresponding areas (in square meters) enclosed by those contours. The contours and their respective areas are as follows:
>
> • Contour 660 m: Area = 5200 m²
> • Contour 665 m: Area = 9400 m²
> • Contour 670 m: Area = 16300 m²
> • Contour 675 m: Area = 22400 m²
> • Contour 680 m: Area = 40700 m²
> • Contour 685 m: Area = 61500 m²
> • Contour 690 m: Area = 112200 m²
> • Contour 695 m: Area = 198100 m²
> • Contour 670 m: Area = 272400 m²
>
> The problem asks for the volume of water when the reservoir level reaches 700 m, using the prismoidal formula. The options for the calculated volume are:
>
> (A) 29.193 × 10⁶ m³
> (B) 28.893 × 10⁶ m³
> (C) 28.493 × 10⁶ m³
> (D) 29.293 × 10⁶ m³
> (E) 30.393 × 10⁶ m³
> (F) 30.093 × 10⁶ m³
> (G) 29.493 × 10⁶ m³
> (H) 29.893 × 10⁶ m³
> (I) 29.793 × 10⁶ m³
> (J) 29.693 × 10⁶ m³
> ************************************************************************************
>
> In this case, the model is extracting the information out from the image that contains the answer choices rather than giving a final answer.

---

> ### Author Response · Authors · 2025-11-21
> **Response to Question 3: language-shortcuts**
>
> We think there is a misunderstanding of what “language shortcut” actually means. A language shortcut occurs when the model overlooks the visual input and answers purely based on textual priors or common-sense biases in the question. For example, a language model might assume that a human hand has five fingers, even if the image actually shows a genetically mutated hand with different numbers.
>
> So, our method can help reduce the language shortcut. In the two-stage method, the visual perception stage forces the VLM to extract factual visual evidence first — such as actually counting the fingers in the image — before the answer generation stage. Because the model is required to commit to a vision-grounded representation before any language-only reasoning, it cannot simply fall back on those textual shortcuts.
>
> We included a detailed analysis of language shortcut in the paper section 3.3.3, and we will also add more explanations in the introduction part to make the concept more clear.

---

### Official Review · Reviewer_BBYi · 2025-11-02

**Soundness:** 2
**Presentation:** 3
**Contribution:** 2
**Rating:** 4
**Confidence:** 4

**Summary:**

This paper proposes Vision-SR1, a self-rewarding reinforcement learning framework that decomposes vision-language reasoning into visual perception (i.e., visual descriptions) and language reasoning stages. The method introduces Multi-Reward Policy Optimization, which separately computes and combines visual perception rewards and answer accuracy rewards, avoiding entangled learning signals. Besides, Vision-SR1 generates self-contained visual perceptions and verifies them using the model itself, improving grounding and reducing hallucination. Experiments on 47K examples show gains across visual reasoning, math, and hallucination benchmarks, outperforming Vision-R1 and related baselines.

**Strengths:**

I believe the studied direction is important -- overcoming the VLMs overly relying on learned priors, particularly text priors. Also, the math equation is descried clearly in the paper.

Empirical results on a couple of comprehensive benchmarks are good, demonstrating improvements. It mostly contains general visual understanding and multimodal mathematical reasoning. Some ablation studies are included as well. It also curated Vision-SR1-47K from some open source VLM benchmarks (table 1), while I am unsure if data may pollute the test set or not, since the performance of finetuning over on this data improve the performance quite a lot --- Vision-R1 47K data (fair comparisons).

**Weaknesses:**

From my perspective, the proposed two-stage pipeline is a bit ad-hoc due to it explicitly enforces there could be two stages, where the first stage generate some visual descriptions, and then the second stage is doing the reasoning. Such pipeline is expectedly not the ultimate goal for VLMs. It is basically a captioning models + a LLM --> this is not something we want to achieve for VLMs. Such behavior should emerge in VLM inference process, not enforced by constructing a small scale datasets with specific setting ⟨visual reasoning⟩ c ⟨/visual reasoning⟩.

For evaluation,I would recommend to include some benchmarks directly motivated by language shortcuts and text priors, such as Probing Visual Language Priors in VLMs and Winoground: Probing vision and language models for visio-linguistic compositionality.

Also, extend to larger scale of models, such as 72B, will make the whole evaluation solid. Extending the experimental results to other base-model beyond Qwen-2.5-VL is also helpful.

**Questions:**

Could the author consider the experimental questions shown above, particularly I curious how the finetuned model perform in the datasets which directly motivated by VLM overly relying on language priors? I will consider the response and other reviewers' comments to adjust my final score, but plz feel free to skip any experiments you believe is unreasonable.

Regarding the two stage things, I will definitely discuss with other reviewers and AC.

---

> ### Author Response · Authors · 2025-11-21
> **Response to questions about two-stage reward process (weakness 1)**
>
> We thank the reviewer for the insightful and constructive feedback. We agree that an ideal VLM should implicitly learn to reason over visual inputs without relying on an explicitly enforced two-stage structure.Our design does not aim to artificially impose a captioning + reasoning split. It serves as a controlled mechanism to analyze and improve visual grounding in reasoning processes, which we find to be inadequately captured in current end-to-end VLM RL training.
>
> Recent works in reinforcement learning post-training (e.g., Vision-R1, LMM-R1, MM-RLHF) have indeed showed that encouraging chain-of-thought (CoT) reasoning improves accuracy and reasoning quality. However, as shown in these studies, such post-training often leads to language-dominant behaviors—the model relies on its linguistic priors to think and guess the answer while its visual grounding weakens over time when more tokens are generated. This phenomenon is sometimes described as less seeing or visual forgetting, where the VLM becomes a strong text reasoner but a weak visual reasoner. [1]
>
> Our proposed training aims to break down the RL training into two stages and assign rewards to enforce visual captioning and final answer to prevent hallucinations and language shortcuts occurring while improving reasoning abilities in VLMs during RL training. Our results show how explicit grounding impacts reasoning quality.
>
> [1] More Thinking, Less Seeing? Assessing Amplified Hallucination in Multimodal Reasoning Models: https://arxiv.org/abs/2505.21523

---

> ### Author Response · Authors · 2025-11-21
> **Response to adding additional language shortcut datasets and models**
>
> We extend our testing benchmarks to three additional datasets that test language short-cut and 3D spatial understanding (also mentioned by reviewer [mWSw](https://openreview.net/forum?id=C1M4ETatgM&noteId=SMZej0W8dg). Additionally, since we have compute limitations, we will only be able to run Qwen2.5-VL-32B Lora finetuning and we will update the results once we have them. But we use the same approach and trained the model on a model that is not in the Qwen-VL family– Mimo-VL-7B [2]
>
> We used the following additional datasets:
>
> ViLP: Probing Visual Language Priors in VLMs and Winoground [3]
>
> MMSI-Bench: A Benchmark for Multi-Image Spatial Intelligence [4]
>
> OmniSpatial: Towards Comprehensive Spatial Reasoning Benchmark for Vision Language Models [5]
>
> ## Pipeline method on models other than Qwen-VL family
>
> First, we evaluate our pipeline method compared to general RL CoT training and show the results in our paper. We will provide the Qwen–VL-32B once they are available.
>
>
> | Model                  | MMMU-Pro | MMMU | ReadWorldQA | VisNumBench | MathVerse | Math-Vision | Hallusionbench | Avg  |
> | ---------------------- | -------- | ---- | ----------- | ----------- | --------- | ----------- | -------------- | ---- |
> | Mimo-7B-Base           | 38.0     | 45.6 | 68.2        | 30.2        | 35.5      | 21.6        | 71.9           | 44.4 |
> | Mimo (Vision-R1)       | 38.7     | 47.3 | 67.1        | 33.5        | 35.3      | 25.7        | 74.3           | 46.0 |
> | Mimo (Vision-SR1 ours) | 39.3     | 49.5 | 68.1        | 44.6        | 40.0      | 29.6        | 75.6           | 49.5 |
>
> ## We show the results of Qwen2.5-VL-7B and Mimo-7B on these language short-cut and spatial intelligence datasets.
>
> | Model                  | ViLP (language priors) | MMSI-Bench | OmniSpatial | Avg  |
> | ---------------------- | ---------------------- | ---------- | ----------- | ---- |
> | Mimo-7B-Base           | 56.4                   | 28.2       | 40.3        | 41.6 |
> | Mimo (Vision-R1)       | 58.2                   | 27.7       | 40.4        | 42.1 |
> | Mimo (Vision-SR1 ours) | 59.3                   | 28.0       | 42.7        | 43.3 |
> | Qwen-7B-Base           | 45.1                   | 24.0       | 27.3        | 32.1 |
> | Qwen (Vision-R1)       | 51.3                   | 21.9       | 31.1        | 34.8 |
> | Qwen (Vision-SR1 ours) | 52.6                   | 27.7       | 44.2        | 41.5 |
>
> Scene understanding is highly vision-intensive and therefore less susceptible to language priors that might otherwise allow the model to guess the correct answer without truly perceiving the visual content.
>
> We will include our new results on Qwen-VL-32B once it is available.
>
> [2] Mimo-VL Technical Report: https://arxiv.org/abs/2506.03569
>
> [3] ViLP: Probing Visual Language Priors in VLMs and Winoground https://arxiv.org/pdf/2501.00569
>
> [4] MMSI-Bench: A Benchmark for Multi-Image Spatial Intelligence https://arxiv.org/abs/2505.23764
>
> [5] OmniSpatial: Towards Comprehensive Spatial Reasoning Benchmark for Vision Language Models https://arxiv.org/abs/2506.03135

---

> ### Comment · Reviewer_BBYi · 2025-11-24
> **Reviewer response**
>
> I appreciate the additional experiments — they strengthen the submission. I wonder whether the authors could provide a revised version that includes these updated results during the rebuttal period. Also, is there any chance you could run the experiments on the 72B model?
>
> Once these updates are incorporated, I will take another look at the revised version and provide a final recommendation to the AC.
>
> I’m prepared to raise my score to a 6 pending the revised submission. The system currently isn’t allowing me to update the score, but I will adjust it as soon as it becomes available.

---

> ### Author Response · Authors · 2025-11-28
> **Response to Reviewer**
>
> We thank the reviewer for their reviews and responses. We have updated the paper PDF submission to reflect all the new results and changes. Our results include:
>
> - Added two spatial reasoning and one extra language shortcut (ViLP) validation datasets stated in our rebuttal responses (addressed in previous comments).
>
> - Added an additional VL model (Mimo-VL-7B) to show generalization of our method other than Qwen-VL model family (addressed in previous comments).
>
> - Added Qwen2.5-VL-72B Lora results to show that our method also generalizes to larger scales of models below and in the updated submission.
>
> - Qwen2.5-VL-72B Lora finetune results on general vision datasets in our original submission:
>
> | Model | MMMU-Pro | MMMU | RealWorldQA | VisNumBench | MathVerse | Math-Vision | HallusionBench | Avg |
> |-------|----------|------|-------------|-------------|-----------|-------------|----------------|-----|
> | Qwen2.5-VL-72B-Base | 40.6 | 45.0 | 69.5 | 26.1 | 51.3 | 33.5 | 68.7 | 47.8 |
> | Qwen2.5-VL-72B (Vision-R1) | 43.8 | 45.3 | 72.1 | 47.1 | 50.5 | 34.6 | 73.2 | 52.4 |
> | Qwen2.5-VL-72B (Vision-SR1 ours) | 47.6 | 52.8 | 75.1 | 47.9 | 53.6 | 34.5 | 74.4 | 55.1 |
>
>
> - Qwen2.5-VL-72B Lora finetune results on three new datasets:
>
> | Model | ViLP (LS) | MMSI-Bench | OmniSpatial | Avg |
> |-------|-----------|------------|-------------|-----|
> | Qwen2.5-VL-72B-Base | 55.9 | 33.4 | 36.5 | 41.9 |
> | Qwen2.5-VL-72B (Vision-R1 Lora) | 55.4 | 35.4 | 36.4 | 42.4 |
> | Qwen2.5-VL-72B (Vision-SR1 Lora ours) | 61.8 | 35.3 | 38.6 | 45.2 |
>
> We believe we have addressed all concerns mentioned.

---

> > ### Author Response · Authors · 2025-11-29
> > **Added new results on the new submission**
> >
> > We thank the reviewer for their critical reviews. Since the 72B model takes a long time to train and run inference, we just got the results and updated the submission PDF to incorporate the changes and concerns.

---

### Meta-Review · Area_Chair_j4Xy · 2026-01-13

**Summary:**

The paper introduces Vision-SR1, a reinforcement-learning post-training framework for LMs that targets visual hallucinations and language shortcuts by explicitly separating (i) visual perception generation and (ii) language reasoning / answer generation conditioned only on that perception. Training uses a multi-reward / multi-loss formulation that computes and combines visual-reasoning reward and answer-accuracy reward in a decoupled manner, aiming to provide denser supervision over intermediate visual reasoning than final-answer-only RL.
THIS submission makes a credible, well-motivated contribution to post-training  RL for VLMs by turning intermediate “visual perception” into a trainable, reward-bearing object without external annotation, and it backs the claims with substantial empirical work and rebuttal-driven strengthening. Sufficient materials in rebuttal address the main concerns (novelty, circularity, cost).

**Reviewer Concerns:**

1. the paper should more explicitly separate what is structurally similar from what is methodologically new and provide direct apples-to-apples baselines where feasible.
2. while CLIP similarity and LSR are helpful, readers want clearer guidance on when self-reward may fail  and how robust the approach is under distribution shift.

**Reviewer Scores:**

most discussion is  the authors directly addressed borderline concerns by expanding evidence in the rebuttal, which appears to have materially increased reviewer confidence, with at least one reviewer indicating willingness to raise a score contingent on incorporation of updated results into the submission.

---

### Decision · Program_Chairs · 2026-01-26

Accept (Poster)